# Toxicity and Physiological Effects of Nine *Lamiaceae* Essential Oils and Their Major Compounds on *Reticulitermes dabieshanensis*

**DOI:** 10.3390/molecules28052007

**Published:** 2023-02-21

**Authors:** Xi Yang, Chunzhe Jin, Ziwei Wu, Hui Han, Zhilin Zhang, Yongjian Xie, Dayu Zhang

**Affiliations:** 1College of Advanced Agricultural Sciences, Zhejiang A and F University, Hangzhou 311300, China; 2Hubei Key Laboratory of Quality Control of Characteristic Fruits and Vegetables, Hubei Engineering University, Xiaogan 432000, China

**Keywords:** termites, *Reticulitermes dabieshanensis*, essential oil, insecticidal, detoxification enzyme

## Abstract

The volatile metabolites of *Salvia sclarea*, *Rosmarinus officinalis*, *Thymus serpyllum*, *Mentha spicata*, *Melissa officinalis*, *Origanum majorana*, *Mentha piperita*, *Ocimum basilicum* and *Lavandula angustifolia* were determined by gas chromatography–mass spectrometry. The vapor insecticidal properties of the analyzed essential oils and their compounds were screened using *Reticulitermes dabieshanensis* workers. The most effective oils were *S. sclarea* (major constituent linalyl acetate, 65.93%), *R. officinalis* (1,8-cineole, 45.56%), *T. serpyllum* (thymol, 33.59%), M. spicata (carvone, 58.68%), *M. officinalis* (citronellal, 36.99%), *O. majorana* (1,8-cineole, 62.29%), *M. piperita* (menthol, 46.04%), *O. basilicum* (eugenol, 71.08%) and *L. angustifolia* (linalool, 39.58%), which exhibited LC_50_ values ranging from 0.036 to 1.670 μL/L. The lowest LC_50_ values were recorded for eugenol (0.060 μL/L), followed by thymol (0.062 μL/L), carvone (0.074 μL/L), menthol (0.242 μL/L), linalool (0.250 μL/L), citronellal (0.330 μL/L), linalyl acetate (0.712 μL/L) and 1,8-cineole (1.478 μL/L). The increased activity of esterases (ESTs) and glutathione S-transferase (GST) were observed but only alongside the decreased activity of acetylcholinesterase (AChE) in eight main components. Our results indicate that *S. sclarea*, *R. officinalis*, *T. serpyllum*, *M. spicata*, *M. officinalis*, *O. marjorana*, *M. piperita*, *O. basilicum* and *L. angustifolia* essential oils (EOs) and their compounds, linalyl acetate, 1,8-cineole, thymol, carvone, citronellal, menthol, eugenol and linalool could be developed as control agents against termites.

## 1. Introduction

Termites are significant agricultural and forestry pests across the world and can seriously threaten the survival of plants and buildings [1]. According to statistics, there are more than 2800 recorded termite variants in the world, 185 of which are considered pests [2]. They cause global economic losses of more than USD 40 billion annually [3]. There is no doubt that chemical pesticides are some of the most effective and widely used methods for termite control [3]. However, the excessive use of pesticides has led to a series of problems, such as the development of insect resistance, ecological imbalance and harm to mammalian and human health [4].

Lamiaceae are annual or perennial herbs or shrubs, which include 10 subfamilies, 236 genera and more than 7000 species [5]. They are mainly distributed in Asia, Europe and Africa. There are more than 99 genera and more than 808 species in China, which are distributed throughout the country, with higher numbers found in the southwest and south. Lamiaceae plants are famous for their rich aromatic oils, many of which can be used for medicine. In particular, the genus *Mentha* possesses anti-inflammatory, anti-emetic, antispasmodic, analgesic, anticancer, anti-obesity, antidiabetic, anti-bloating, and immunomodulatory actions [6].

Most Lamiaceae EOs contain rich amounts of volatile components, which function as fumigators, antifeedants and repellents and display contact toxicity and inhibit growth and reproduction of pests. *Lavandula angustifolia* EO can control *Rhyzopertha dominica* through fumigation [7] and *Ectropis obliqua* hypulina [8] and *Thrips tabaci* [9] through antifeedant action. The EOs of *Ocimum basilicum* and *O. gratissimum* can prevent and control *Callosobruchus macrotus*, *Oryzaephilus suramensis*, *Acanthoscelides obtectus* and *Tetranychus urticae* Koch [6,10,11,12] through fumigation. *L. angustifolia* and *L. latifolia* EOs have toxicity and repellent effects on adult *Tetranychus cinnabarinus* [13]. *Thymus serpyllum* EO showed good contact and fumigation activity against *Myzus persicae* and *Acanthoscelides obtectus* [14,15]. *Ocimum basilicum* EO can inhibit the oviposition of *Tetranychus cinnabarinus* [13,16]. The EOs of *O. basilicum* and *O. gratissimum* have a strong inhibitory effect on the egg hatching and larval development of the *Callosobruchus maculatus* [11]. *Rosmarinus officinalis* EO is an oviposition deterrent against *A. obtectus* and *E. obliqua* hypulina, and its oviposition deterrent rate for *A. obtectus* can reach 92.0% [9,17].

However, there are almost no reports on the fumigant efficacy of Lamiaceae species EOs against *Reticulitermes dabieshanensis*. Thus, the objective of the present study was (1) to evaluate the fumigant activities of *Salvia sclarea*, *Rosmarinus officinalis*, *Thymus serpyllum*, *Mentha spicata*, *Melissa officinalis*, *Origanum majorana*, *Mentha piperita*, *Ocimum basilicum* and *Lavandula angustifolia* EOs; (2) to investigate eight kinds of EOs’ constituents; and (3) to determine the activities of detoxification enzymes and acetylcholine esterase.

## 2. Results

### 2.1. GC–MS Analysis

The chemical compositions of Lamiaceae EOs are shown in Table 1. The major constituent of *S. sclarea* is linalyl acetate (65.93%), and the main component in *R. officinalis* is 1,8-cineole, where the content is 45.56%. Thymol (33.59%) is the main component of *T. serpyllum*. The major component of *M. spicata* is carvone (58.68%). The main component detected in *M. officinalis* was citronellal (36.99%). 1,8-Cineole (62.29%) was identified as a major component of *O. majorana* oil. The most abundant component in *M. piperita* oil was menthol (46.04%), and eugenol (71.08%) was the most abundant in *O. basilicum* oil. The main component of *L. angustifolia* is linalool, with the content of 39.58%.

### 2.2. Fumigation Activity of Lamiaceae EOs and Its Major Constituents

According to Table 2, the LC_50_ values of *O. basilicum*, *M. spicata*, *T. serpyllum*, *M. piperita*, *M. officinalis*, *L. angustifolia*, *S. sclarea*, *O. majorana* and *R. officinalis* EOs against *R. dabieshanensis* were 0.048, 0.060, 0.137, 0.321, 0.564, 0.690, 1.015, 1.029 and 1.904 μL/L, respectively. 

The fumigation activity of the major components was further determined, and the results are shown in Table 3. Among the eight components tested, those with the highest toxicity were eugenol (LC_50_ = 0.060 μL/L), followed by thymol (LC_50_ = 0.062 μL/L), carvone (LC_50_ = 0.074 μL/L), menthol (LC_50_ = 0.242 μL/L), linalool (LC_50_ = 0.250 μL/L), citronellal (LC_50_ = 0.330 μL/L), linalyl acetate (LC_50_ = 0.712 μL/L) and 1,8-cineole (LC_50_ = 1.478 μL/L).

### 2.3. ESTs, GST and AChE Enzyme Activities

As compared with the control, treatment with linalyl acetate, 1,8-cineole, thymol, carvone, citronellal, menthol, eugenol and linalool demonstrated increased activities of esterase (for α-NA, *F* = 97.816, *d.f.* = 8,18, *p* < 0.0001; for β-NA, *F* = 239.570, *d.f.* = 8,18, *p* < 0.001). However, carvone (α-NA) and thymol (β-NA) showed the highest esterase activity in all treatments (Table 4). The activity of GST also significantly increased in *R. dabieshanensis* through exposure to linalyl acetate, 1,8-cineole, thymol, carvone, citronellal, menthol, eugenol and linalool compared with the control (*F* = 64.099, *d.f.* = 8,18, *p* < 0.001) (Table 4). On the other hand, in all treatments, the activity of acetylcholinesterase was significantly decreased (*F* = 50.467, *d.f.* = 8,18, *p* < 0.001), and of the test oils and compounds, eugenol showed the highest inhibition activity.

Table 5 summarizes the inhibitory effects of eight major constituents on AChE activity. The IC_50_ of 1,8-cineole, linalool, eugenol, linalyl acetate, carvone and thymol were estimated to be 0.097, 0.136, 0.501, 0.601, 1.922 and 6.360 μL/mL, respectively (Table 5). Other than through citronellal and menthol, there was no significant inhibition on the acetylcholinesterase activity of *R. dabieshanensis*.

## 3. Discussion

The present study found that the main components of the nine Lamiaceae EOs were linalyl acetate, 1,8-cineole, thymol, carvone, citronellal, menthol, eugenol and linalool, which were consistent with the main components of the EOs studied by Tuttolomondo et al. [18], Apostolides et al. [19], Kim et al. [20], Park et al. [21], Mafakheri et al. [22], Krasnewska et al. [23], Goudarzian et al. [24], Raina et al. [25] and Kara and Baydar [26].

In our study, strong insecticidal activity against *R. dabieshanensis* was achieved with essential oils of *S.sclarea*, *R. officinalis*, *T. serpyllum*, *M. spicata*, *M. officinalis*, *O. majorana*, *M. piperita*, *O. basilicum* and *L. angustifolia,* with the LC_50_ values of 0.060–1.478 μL/L. These results agree with those of Xie et al. [27], who demonstrated the antitermitic activity of *Syzgium aromaticum* EO against *R. chinensis* (LC_50_ = 12.5 μg/g) after 7 d. Similarly, Pandey et al. [28] have also reported the antitermitic activity of *S. aromaticum* EO on *Odontotermes assamensis*. Yang et al. [1] have recently demonstrated that the LC_50_ value of spearmint EO against *R. dabieshanensis* was 0.194 μL/L. Jin et al. [29] showed that lemongrass EO had high toxicity against *R. flaviceps* (LC_50_ = 0.328 μL/L). 

There are no previous studies on the insecticidal activities of Lamiaceae EOs against *R. dabieshanensis*; however, there have been previous reports on the insecticidal potential of Lamiaceae EOs. Koliopoulos et al. [30] reported that *Mentha spicata*, *M. longifolia*, *M. suaveolens*, *Melissa officinalis*, *Salvia fruticosa*, *S. pomifera* subsp. calycina and *S. pomifera* subsp. pomifera revealed larvicidal activity against *Culex pipiens* with LC_50_ values ranging from 47.88 to 91.45 mg/L. Papachristos and Stamopoulos [17] demonstrated the adulticidal activity of *Rosmarinus officinalis* EO against the males (LC_50_ = 2.1 μL/L) and females (LC_50_ = 3.3 μL/L) of *Acanthoscelides obtectus*. Similarly, Sertkaya et al. [31] reported that *Thymus serpyllum* EO (1.12 µg/mL), followed by *Origanum onites* (1.31 µg/mL), *Rosmarinus officinalis* (2.66 µg/mL), *Ocimum basilicum* (3.10 µg/mL) and *Melissa officinalis* (3.60 µg/mL), respectively, displayed high adulticidal activity against the bean weevil adult, *Acanthoscelides obtectus* (Say). Štefanidesová et al. [32] also found that *Thymus serpyllum* essential oil repelled 82% of *Dermacentor reticulatus* adults when diluted to 3%. The colonization rate of *Myzus persicae* was as low as 10.0% after being treated with the essential oil of *Mentha spicata* for 6 h [14]. Koudal et al. [33] demonstrated that *M. piperita* had significant toxicity against *Plutella xylostella* (LC_50_ = 1.37 mg/mL). Yarou et al. [16] found that *Ocimum gratissimum* and *O. basilicum* significantly reduced *Tuta absoluta* oviposition behavior on a tomato plant. These studies indicate that Lamiaceae EOs have broad application prospects in pest control. Similarly, this study found that Lamiaceae EOs have good control effects on termites, further proving that Lamiaceae EOs can play a huge role in pest control. 

To explore the relationship between the constituents of plant EOs and termiticidal activity, eight main components were tested for insecticidal activity against *R. dabieshanensis*. In this study, linalyl acetate, 1,8-cineole, thymol, carvone, citronellal, menthol, eugenol and linalool displayed effective vapor activity against *R. dabieshanensis*, which are the major components of the nine selected EOs. In general, the insecticidal activity of the EOs may be attributed to their major component, as has also been reported in some previous studies [1,29,34,35]. Here, the *O. basilicum* EO showed the highest insecticidal activity in comparison with its major constituent, eugenol, against *R. dabieshanensis*. Similarly, Piri et al. [36] found that the Ajwain EO showed the highest insecticidal activity in comparison with its constituents against *Tuta absoluta* larvae. Shahriari et al. [37] reported that the Ajwain EO was more toxic to *Ephestia kuehniella* larvae than thymol. Results from this study suggested that the EOs exhibited termiticidal activity which can be attributed to their major active chemical constituents.

EOs comprise lipophilic and low-molecular-weight volatile compounds, with terpenoids and phenylpropanoids as the most common constituents. Our results demonstrate that the linalyl acetate, 1,8-cineole, thymol, carvone, citronellal, menthol, eugenol and linalool display effective vapor activity against *R. dabieshanensis*. Previously, monoterpenes were found to possess varying insecticidal activities on the various insect species [38,39,40]. From the results of the present study, it is expected that monoterpenes will be able to be used successfully as a control agent against *R. dabieshanensis*.

In addition, it is known in the literature that most of the EOs and their major components can exert their toxic efficacy on insects, notably through inhibition of P450 cyto-chromes (CYPs) [41], GABA receptors [42], octopamine synapses [43], tyramine receptors [44] and the inhibition of acetylcho-linesterase (AchE) [1]. Furthermore, these components from various plant kingdoms can also regulate the intracellular pathways of mitochondrial biogenesis, through the removal of damaged mitochondria (mitophagy) and the generation of new ones required to preserve the cellular and mitochondrial homeostasis [45]. 

To further explore the physiological effect of Lamiaceae EOs on *R. dabieshanensis*, the changes of two detoxification enzymes (esterase, glutathione transferase), one hydrolase (acetylcholinesterase) and the activity of acetylcholinesterase in vitro in *R. dabieshanensis* were measured. The results in Table 4 show that the activities of esterase and glutathione transferase increase and the activities of acetylcholinesterase decrease after the termites are treated with the main ingredients. With the increase in concentration, the inhibitory activity of acetylcholinesterase in vitro also increased. These studies indicate that the essential oil of Lamiaceae may lead to the death of *R. dabieshanensis* by inhibiting the activity of acetylcholinesterase.

Shahriari et al. [37], Piri et al. [36], Wang et al. [46] and Yang et al. [1] found that after treatment with an essential oil or its components, the activities of ESTs and GST of insects increased significantly, indicating that ESTs and GST may participate in the detoxification process of insects. Table 4 shows that the activities of ESTs and GST of termites after treatment are significantly increased. In addition, studies have shown that essential oils and their main components can produce toxic effects on insects by inhibiting acetylcholinesterase (AchE) [36,46]. For instance, carvone showed the effect of inhibiting acetylcholinesterase (70.20% at 0.05 M) in *Tribolium castaneum* [47], while dihydrocarvone showed strong acetylcholinesterase inhibitory activity (IC_50_ = 1.60 mg/mL) in *Blattella germanica* [48].

Our results indicate that *S. sclarea*, *R. officinalis*, *T. serpyllum*, *M. spicata*, *M. officinalis*, *O. majorana*, *M. piperita*, *O. basilicum* and *L. angustifolia* EOs and their compounds could be developed as control agents against termites. For the practical use of these oils and their constituents as novel termite-control agents, the safety of the oils and their compounds in humans and nontarget organisms and their modes of action should be investigated further.

## 4. Materials and Methods

### 4.1. Plant EOs and Their Constituents

*Salvia sclarea*, *Rosmarinus officinalis*, *Thymus serpyllum*, *Mentha spicata*, *Melissa officinalis*, *Origanum majorana*, *Mentha piperita*, *Ocimum basilicum* and *Lavandula angustifolia* EOs were purchased from Shanghai Zixin Biotechnology Co., Ltd. Linalyl acetate, 1,8-cineole, thymol, carvone, citronellal, menthol, eugenol, linalool and other main ingredients were purchased from Shanghai Sigma–Aldrich Trading Co., Ltd.

### 4.2. Termites

Three colonies of *R. dabieshanensis* were collected from Linglong Mountain Scenic Area, Lin’an District, Hangzhou City, Zhejiang Province (longitude 30.2251° N, latitude 119.6843° E), and reared with water and newspapers in a laboratory. The healthy and active termite workers of uniform size were selected for further experiments.

### 4.3. GC–MS Analysis

The chemical analyses of EOs were determined by GC–MS. A gas chromatograph (Agilent 6890A, Santa Clara, CA, USA) was used with an HP-5MS capillary column (30 m × 0.25 mm i.d., 0.25 μm film thickness). The flow rate of helium carrier gas was set at 1.0 mL/min, the split ratio was set at 1:50 and a sample volume of 1.0 μL was injected. The injector and detector temperatures were set at 250 °C. The mass range was scanned from 15 to 500 *m*/*z*. The compound composition was identified by comparing its retention index with the NIST11.LIB database and the Adams [49] library.

### 4.4. Fumigant Toxicity

In order to conduct fumigations [36], filter paper strips (1.5 × 6 cm) were stuck to the lids of 1 L glass jars (10 cm diameter × 12.5 cm), and 0.04–3.0 μL of nine EOs, their major components or acetone as a control was added. Twenty healthy workers were put into a glass bottle, the bottle cap was quickly closed and a moist filter paper was placed on the bottom of the bottle as food. The experiment was repeated three times with three colonies, and the glass jars were kept at 25 ± 1 °C and 75 ± 5% RH. After 24 h, the number of dead termites was recorded.

### 4.5. Determination of Enzyme Activity

#### 4.5.1. Enzyme Assays

The effects of major constituents on the esterase enzymes, glutathione S-transferase and acetylcholine esterase against the worker adults of *R. dabieshanensis* were determined at the LC_30_ concentrations. Enzyme extracts were prepared from five termite workers, homogenized in 1 mL 0.1 M phosphate buffer (pH 7.0) and centrifuged at 4 °C and 12,000× *g* for 15 min; then, the supernatants were placed in a 1.5 mL microcentrifuge tube and stored at −80 °C for later use.

#### 4.5.2. Esterase (EST) 

EST activity was determined utilizing the method of Yang et al. [1]. A total of 20 μL of 10 mM α-naphthyl acetate (α-NA) and β-naphthyl acetate (β-NA) was added separately, and, after that, 10 μL enzyme solution and 50 μL of 1 mM fast blue RR Salt were added. After mixing for 5 min at 27 °C, the OD value was measured at 450 nm with a 96-well microplate reader. 

#### 4.5.3. Glutathione S-Transferase (GST)

The GST activity was determined according to the method of Yang et al. [1]. The reaction solution contained 20 μL of 20 mM 1-chloro-2,4-dinitrobenzene (CDNB) and 10 μL of enzyme solution. After incubation at 27 °C for 5 min, the OD value was measured at 340 nm using a 96-well microplate reader. 

#### 4.5.4. Acetylcholinesterase (AChE)

Acetylcholinesterase activity was determined using the method of Yang et al. [1]. The reaction solution was incubated at 25 °C for 5 min and contained 80 μL 0.1 M phosphate buffer (pH 7.0), 50 μL 10 mM acetylcholine iodide and 50 μL 10 mM of 5,5-dithiobis-2- nitrobenzoic acid (DTNB), which was then added to 20 μL of enzyme solution. The OD value was measured at 405 nm using a 96-well microplate reader.

#### 4.5.5. Acetylcholinesterase Inhibition 

In an AChE inhibition test, five termites were ground using a porcelain mortar in 0.1 M Tris-HCl buffer (pH 7.8) (0.02 M NaCl and 0.5% Triton X-100). Then, the ground termites were centrifuged at 15,000× *g* for 15 min at 4 °C. The reaction solution contained 20 μL of the tested compound, 40 μL of enzyme solution, 50 μL of 10 mM acetylthiocholine iodide, 10 μL 4 mM DTNB and 100 μL of protein extraction buffer. After incubation at 27 °C for 30 min, the OD value was measured at 412 nm using a 96-well microplate reader.

### 4.6. Data Analysis

Toxicity data were subjected to probit analysis in order to estimate the LC_50_ values of nine EOs, their major constituents and 50% inhibition AChE activity (IC_50_). The data of the mortality and inhibition rates were analyzed by one-way ANOVA and Duncan’s multiple comparison method, with a significance level of *p* < 0.05.

## Figures and Tables

**Table 1 molecules-28-02007-t001:** Chemical constituents of nine essential oils of Lamiaceae.

No	Components	RI	Relative Percentage Content (%)								
			1	2	3	4	5	6	7	8	9
1	α-Pinene	939	-	23.92	0.64	-	-	6.60	0.65	-	-
2	Camphene	954	-	4.72	-	-	-	-	-	-	-
3	β-Pinene	979	-	4.86	2.03	-	-	2.58	1.91	-	-
4	β-Myrcene	991	-	-	-		-	-	-	-	1.98
5	β-Phellandrene	1001	2.95	-	0.36	-	-	0.73	-	-	-
6	α-Terpinene	1018	-	2.18	-	-	-	-	-	-	-
7	p-Cymene	1025	-	-	28.32	-	-	-	-	-	-
8	Limonene	1027	-	-	-	21.28	3.81	-	5.65	-	6.34
9	1,8-Cineole	1038	-	45.56	-	-	-	62.29	-	-	-
10	β-Ocimene	1046	1.59	-	-	-	-	1.06	-	-	2.05
11	γ-Terpinene	1060	-	0.91	31.02	-	-	1.31	-	-	-
12	Linalool	1097	17.57	-	-	-	0.84	15.40	-	2.17	39.58
13	Camphor	1114	0.98	11.33	-	-	-	1.44	-	-	2.57
14	Menthone	1129	-	-	-	1.04	-	-	20.47	-	-
15	Isopulegol	1141	-	-	-	-	-	-	0.96	-	-
16	Isoborneol	1143	-	-	-	-	-	1.21	-	-	0.17
17 18	Citronellal Borneol	1154 1166	- -	- 0.94	- -	- -	36.99 -	- -	- -	- -	- 1.30
19 20	Menthol Neodihydrocarveol	1170 1174	- -	- -	- -	- 11.23	- -	- -	46.04 -	- -	- -
21	Terpinen-4-ol	1177	-	-	-	-	-	1.46	-	-	0.54
22 23	α-Terpineol Estragole	1191 1201	- -	0.47 -	- -	- -	- -	1.42 -	3.30 -	- 18.05	0.44 -
24 25	Citronellol Pulegone	1233 1235	- -	- -	- -	- -	13.77 -	- -	- 1.34	- -	- -
26 27 **28** **29** **30** **31** **32** **33** **34** **35** **36** **37** **38** **39** **40** **41** **42** **43** **44** **45**	Carvone Geraniol Linalyl acetate Bornyl acetate Lavandulyl acetate Thymol Menthyl acetate Terpinyl acetate Neryl acetate Eugenol α-Copaene Geranyl acetate β-bourbonene β-Elemene Caryophyllene β-Farnesene Humulene Germacrene D δ-Cadinene α-elemol Total	1243 1250 1253 1286 1288 1292 1322 1331 1356 1359 1377 1380 1381 1391 1419 1447 1455 1485 1523 1549	- - 65.93 - - - - - 1.59 -- 2.89 3.38 - - 2.08 - - - - - 98.96	- - - 1.59 - - - - - - - - - - 3.02 - - - - - 99.51	- - - - - 33.59 - - - - - - - - - - - - - 95.97	58.68 - - - - - - - - - - - 1.97 0.78 2.21 - - - - - 97.20	- 20.23 - - - - - - - - - 3.29 - 3.49 - - - 1.81 3.59 5.86 93.67	- - - - - - - 0.73 - - - - - - 1.38 - - - - - 97.60	- - - - - - 6.31 - - - - - - - 10.83 1.83 - - - 99.29	- - - - - - - - - 71.08 - - - - 6.73 - 1.57 - - - 99.60	- - - - 1.80 - - - 0.58 - - 1.53 - - 2.95 0.57 - 0.23 - - 97.08

1. *S. sclarea*; 2. *R. officinalis*; 3. *T. serpyllum*; 4. *M. spicata*; 5. *M. officinalis*; 6. *O. majorana*; 7. *M. piperita*; 8. *O. basilicum*; 9. *L. angustifolia*.

**Table 2 molecules-28-02007-t002:** LC_50_ values (μL/L) of nine essential oils from Lamiaceae against *R. dabieshanensis*.

EOs	Con. (μL/L)	Mortality (% ± SD)	LC_30_ (95%CL *)	LC_50_ (95%CL)	LC_90_ (95%CL)	*χ* ^2^
*S. sclarea*	0.16	15.00 ± 8.66	0.604 (0.480–0.726)	1.015 (0.854–1.199)	3.605 (2.812–5.081)	17.571
	0.31	28.33 ± 10.41				
	0.63	55.00 ± 22.91				
	1.25	81.67 ± 7.64				
	2.50	96.67 ± 2.89				
*R. officinalis*	1.00	0.00 ± 0.00	1.670 (1.494–1.805)	1.904 (1.755–2.051)	2.625 (2.391–3.046)	24.728
	1.50	28.33 ± 7.64				
	2.00	36.67 ± 2.89				
	2.50	90.00 ± 10.00				
	3.00	100.00 ± 0.00				
*T. serpyllum*	0.08	26.67 ± 16.07	0.092 (0.066–0.116)	0.137 (0.108–0.166)	0.360 (0.282–0.531)	24.147
	0.16	53.33 ± 15.28				
	0.31	86.67 ± 18.93				
	0.63	98.33 ± 2.89				
	1.25	100.00 ± 0.00				
*M. spicata*	0.04	26.67 ± 7.64	0.043 (0.035–0.051)	0.060 (0.051–0.068)	0.129 (0.109–0.165)	9.890
	0.08	66.67 ± 10.41				
	0.16	95.00 ± 8.66				
	0.31	100.00 ± 0.00				
	0.63	100.00 ± 0.00				
*M. officinalis*	0.16	3.33 ± 5.77	0.425 (0.331–0.516)	0.564 (0.462–0.695)	1.126 (0.880–1.705)	29.770
	0.31	13.33 ± 7.64				
	0.63	46.67 ± 15.28				
	1.25	98.33 ± 2.89				
	2.5	100.00 ± 0.00				
*O. majorana*	0.31	10.00 ± 5.00	0.684 (0.570–0.795)	1.029 (0.890–1.188)	2.799 (2.294–3.647)	10.082
	0.63	18.33 ± 2.89				
	1.25	60.00 ± 5.00				
	2.5	91.67 ± 5.77				
	5	96.67 ± 2.89				
*M. piperita*	0.15	36.67 ± 10.41	0.187 (0.090–0.268)	0.321 (0.209–0.432)	1.209 (0.812–2.809)	34.831
	0.3	38.33 ± 5.77				
	0.6	51.67 ± 10.41				
	0.9	93.33 ± 7.64				
	1.2	96.67 ± 5.77				
*O. basilicum*	0.04	31.67 ± 7.64	0.036 (0.019–0.047)	0.048 (0.032–0.061)	0.096 (0.074–0.173)	37.174
	0.08	93.33 ± 7.64				
	0.16	95.00 ± 8.66				
	0.31	100.00 ± 0.00				
	0.63	100.00 ± 0.00				
*L. angustifolia*	0.16	10.00 ± 0.00	0.444 (0.338–0.551)	0.690 (0.556–0.865)	2.027 (1.492–3.270)	21.971
	0.31	11.67 ± 2.89				
	0.63	41.67 ± 8.93				
	1.25	70.00 ± 13.23				
	2.5	100.00 ± 0.00				

CL *: confidence limit which has been calculated with 95% confidence.

**Table 3 molecules-28-02007-t003:** LC_50_ values (μL/L) of eight main chemical constituents against *R. dabieshanensis*.

Com.	Con. (μL/L)	Mortality (% ± SD)	LC_30_ (95%CL *)	LC_50_ (95%CL)	LC_90_ (95%CL)	*χ* ^2^
Linalyl acetate	0.16	8.33 ± 7.64	0.431 (0.352–0.510)	0.712 (0.605–0.842)	2.435 (1.886–3.446)	10.023
	0.31	20.00 ± 0.00				
	0.63	33.33 ± 2.89				
	1.25	80.00 ± 0.00				
	2.5	90.00 ± 0.00				
1,8-cineole	1	23.33 ± 7.64	1.052 (0.83–1.240)	1.478 (1.256–1.679)	3.392 (2.959–4.063)	13.979
	2	75.00 ± 10.00				
	3	88.33 ± 5.77				
	4	90.00 ± 5.00				
	5	96.67 ± 5.77				
Thymol	0.02	16.67 ± 7.64	0.038 (0.031–0.045)	0.062 (0.054–0.073)	0.209 (0.153–0.355)	12.663
	0.04	21.67 ± 12.58				
	0.06	45.00 ± 5.00				
	0.08	65.00 ± 5.00				
	0.1	71.67 ± 7.64				
Carvone	0.03	13.33 ± 2.89	0.054 (0.046–0.061)	0.075 (0.067–0.083)	0.168 (0.144–0.210)	16.108
	0.06	28.33 ± 7.64				
	0.09	48.33 ± 2.89				
	0.12	81.67 ± 2.89				
	0.15	93.33 ± 7.64				
Citronellal	0.2	26.67 ± 11.55	0.237 (0.162–0.286)	0.330 (0.269–0.387)	0.745 (0.580–1.302)	28.105
	0.3	38.33 ± 28.43				
	0.4	60.00 ± 15.00				
	0.5	66.67 ± 5.77				
	0.6	91.67 ± 2.89				
Menthol	0.04	10.00 ± 13.23	0.138 (0.091–0.189)	0.242 (0.177–0.358)	0.964 (0.577–2.646)	33.504
	0.08	13.33 ± 12.59				
	0.16	21.67 ± 7.64				
	0.31	63.33 ± 10.41				
	0.63	85.00 ± 5.00				
Eugenol	0.04	23.33 ± 7.64	0.044 (0.036–0.050)	0.060 (0.054–0.067)	0.133 (0.114–0.169)	16.685
	0.06	51.67 ± 7.64				
	0.08	71.67 ± 5.77				
	0.1	73.33 ± 16.07				
	0.12	88.33 ± 12.58				
Linalool	0.2	41.67 ± 10.41	0.166 (0.088–0.218)	0.256 (0.183–0.307)	0.739 (0.567–1.372)	19.116
	0.4	51.67 ± 10.41				
	0.6	71.67 ± 10.41				
	0.8	78.33 ± 10.41				
	1.0	86.67 ± 2.89				

CL *: confidence limit which has been calculated with 95% confidence.

**Table 4 molecules-28-02007-t004:** Effects of eight main components on the enzyme activity of *R. dabieshanensis*.

Reagent	ESTs		GST	ATCh
	α-NA	β-NA		
Control	0.422 ± 0.061 f	1.000 ± 0.091 e	35.410 ± 0.682 e	17.710 ± 1.692 a
Linalyl acetate	0.914 ± 0.054 d	1.287 ± 0.057 b	43.465 ± 2.989 d	8.848 ± 1.033 e
1,8-Cineole	1.180 ± 0.063 b	1.404 ± 0.013 a	65.215 ± 3.181 a	6.683 ± 0.649 fg
Thymol	0.760 ± 0.062 e	1.456 ± 0.076 a	38.507 ± 1.226 e	10.948 ± 1.437 d
Carvone	1.749 ± 0.041 a	1.060 ± 0.155 d	47.806 ± 0.796 c	15.038 ± 1.148 b
Citronellal	0.510 ± 0.040 f	1.293 ± 0.073 c	43.917 ± 3.692 d	8.007 ± 0.665 ef
Menthol	0.949 ± 0.094 cd	1.445 ± 0.088 a	52.972 ± 2.106 b	12.465 ± 0.466 cd
Eugenol	1.067 ± 0.084 bc	1.364 ± 0.042 ab	61.590 ± 1.445 a	6.032 ± 0.137 g
Linalool	0.737 ± 0.098 e	1.391 ± 0.078 a	51.813 ± 0.445 b	13.675 ± 0.340 bc
*df*	8	8	8	8
*F*-value	97.816	239.570	64.099	50.467
Pr	0.0001	0.0001	0.0001	0.0001

Activity of ESTs, GST and ATCH for 24 h of major components (LC_30_) treatment; control only treated with acetone. Mean (±SD) values with different letters (a–g) are significantly different at the level of *p* < 0.05 according to Duncan’s test.

**Table 5 molecules-28-02007-t005:** In vitro assay for half-inhibitory concentration (μL/mL) for eight main components.

Reagent	95%CL	*χ*^2^(*df*)
Linalyl acetate	0.601 (0.311–0.881)	33.821 (4)
1,8-Cineole Thymol Carvone Citronellal Menthol Eugenol Linalool	0.097 (0.024–0.203) 6.360 (4.457–11.487) 1.922 (1.131–3.308) -* - 0.501 (0.055–0.978) 0.136 (0.066–0.218)	17.517 (4) 29.602 (4) 12.262 (4) - - 45.778 (4) 7.738 (4)

-*: No detection.

## Data Availability

The data presented in this study are available on request from the corresponding author.

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
