# Peer review of "Toxicity and Physiological Effects of Nine Lamiaceae Essential Oils and Their Major Compounds on Reticulitermes dabieshanensis"

_molecules, 2023, doi:10.3390/molecules28052007_

Round 1

Author Response

Thank you very much for your comments regarding our manuscript. Those comments are very valuable and helpful for improving our paper.

These problems in this paper are listed as follows:

Abstract: Include method briefly. Make the full form of Abbreviated terms.

Response: We changed the abbreviation mentioned for the first time into the full name according to the opinion.

Introduction: It is good.

Response: We appreciate your comments.

Result: Put the name of plant in Table 1 where the essential oils (EOs) are identified. In Table 1, no. 28 is acetate?

Response: We added the species each number corresponds in Table 1 footnote. We confirm that no. 28 is acetate.

Which control compound is used to compared enzyme activity of R. dabieshanensis on purchased essential oil? Put the name of control in place of ck in Table 4.

Response: All samples in our experiment use acetone as solvent, so acetone is used as control.

In table 2, 3 and 5, are the values given in LCs or IC50 in bracket are ranged?

Response: We carefully checked Tables 2, 3 and 5, and the values of LC or IC50 are within the brackets.

Author purchased the essential oils of Lamiaceae family plants and other essential oil from the company. What is the significance of the study if commercially available essential oil is used? Explain the traditional or other uses of those EOs purchased.

Response: The essential oils sold in the market are generally used in medical treatment, health care, food and spices, etc. In recent years, because of the excellent characteristics of plant essential oil in killing insects, researchers are constantly exploring the essential oil resources. Therefore, the purchased EO can quickly evaluate its biological activity.

It would be better if the extracts or compounds isolation from plants are used for the experiments.

Response: The reviewer's opinion is very good. We may separate relatively novel compounds from plants. Generally speaking, especially commercially available essential oil, their components are relatively clear due to more research in other aspects.

Reviewer 2 Report

1.It is suggested to add more in vivo and clinical studies in introduction part.
2.what is the suggestion of this study for future works?
3.Please discuss and compare your results with previous works and add suggestions.
4.It will be better to add the role of mitochondria.
5.Please add role of cell junctions proteins.
7.More references for the discussion part of manuscript and bold your study novelty should be added: e.g.,
-DOI: 10.1016/j.arabjc.2021.103106
-DOI: 10.1155/2021/4946711

Author Response

Thank you very much for your comments regarding our manuscript. Those comments are very valuable and helpful for improving our paper.

These problems in this paper are listed as follows:

  1. It is suggested to add more in vivo and clinical studies in introduction part.

Response: We added in vivo and clinical research in the introduction. eg. In particular, the genus Mentha possess anti-inflammatory, anti-emetic, antispasmodic, analgesic, anticancer, anti-obesity, antidiabetic, anti-bloating, and immunomodulatory actions.

  1. what is the suggestion of this study for future works?

Response: Our results indicate that S. sclarea, R. officinalis, T. serpyllum, M. spicata, M. officinalis, O. majorana, M. piperita, O. basilicum and L. angustifolia essential oils (EOs) and their compounds could be developed as control agents against termites.

  1. Please discuss and compare your results with previous works and add suggestions.

Response: The reviewer's opinion is very good. We added, In our study, strong insecticidal activity against R. dabieshanensis were achieved with essential oils of S.sclarea, R. officinalis, T. serpyllum, M. spicata, M. officinalis, O. majorana, M. piperita, O. basilicum and L. angustifolia, with the LC50 values of 0.060-1.478 μL/L. These results agree with those of Xie et al. [2015], who demonstrated the antitermitic activity of Syzgium aromaticum EO against R. chinensis (LC50 = 12.5 μg/g) after 7 d. Similarly, Pandey et al. [2012] have also reported the antitermitic activity of S. aromaticum EO on Odontotermes assamensis. Yang et al. [3] have recently demonstrated that the LC50 value of spearmint EO against R. dabieshanensis was 0.194 μl/L. Jin et al. (2022) showed that lemongrass EO had high toxicity against R. flaviceps (LC50 = 0.328 μl/L).

  1. It will be better to add the role of mitochondria. 5. Please add role of cell junctions proteins.

Response: The reviewer's opinion is very good. We added, In addition, it is known in the literature that most of the EOs and their major com-ponents can exert their toxic efficacy on insects notably through inhibition of P450 cy-to-chromes (CYPs)( Belzile et al., 2000), GABA receptors (Tong et al., 2013), octopamine synapses(Enan 2005), tyramine receptors (Lei et al., 2010) and the inhibition of acetyl-cho-linesterase (AchE) [1]. Furthermore, these components from various plant kingdoms can also regulate the intracellular pathways of mitochondrial biogenesis, through removal of damaged mitochondria (mitophagy) and the generation of new ones are required to preserve the cellular and mitochondrial homeostasis [Chodari et al., 2021].

  1. More references for the discussion part of manuscript and bold your study novelty should be added: e.g.,

-DOI: 10.1016/j.arabjc.2021.103106

-DOI: 10.1155/2021/4946711

Response: The reviewer's opinion is very good. We refer to these references in the discussion section

Reviewer 3 Report

Dear Authors, your manuscript titled " Toxicity and Physiological Effects of Nine Lamiaceae Essential Oils and Their Major Compounds on Reticulitermes dabieshanensis" it is very interesting. The use of essential oils as insecticides is a challange for the next future. But your manuscript presents, in my opinion, some details that can be improved:

1.     In all cases it is Origanum majorana (line 14, 19, 63, 73, 85, 173)

2.     Line 18: Me. officinalis is M. officinalis.

3.     Please mention the abbreviations in full form during their first usage (line 24: ESTs, GST, AChE; line 26 EOs).

4.     Figure 1, is not necessary.

5.     Table 1: in the footnote mention to which species each number corresponds

6.     Line 27 and line 28: please correct the name of the compounds.

7.     Lines 131, 133: Rosmarinus officinalis in italics

8.     What were the reference standards used to determine the retention index? Please add them in the GC-MS analysis section.

9.     Why were essential oils not used in the enzyme activity assay? If the authors could make this determination, they might have more evidence to support their conclusions.

10.  In the discussion section, you could make evident the essential oil that showed the highest insecticidal activity and relate it to its chemical composition.

Author Response

Thank you very much for your comments regarding our manuscript. Those comments are very valuable and helpful for improving our paper.

These problems in this paper are listed as follows:

  1. In all cases it is Origanum majorana (line 14, 19, 63, 73, 85, 173)

Response: We carefully checked the full text and corrected the related words spelling errors one by one.

  1. Line 18: Me. officinalis is M. officinalis.

Response: We carefully checked the full text and corrected the related words spelling errors one by one.

  1. Please mention the abbreviations in full form during their first usage (line 24: ESTs, GST, AChE; line 26 EOs).

Response: We changed the abbreviation mentioned for the first time into the full name according to the opinion.

  1. Figure 1, is not necessary.

Response: We deleted the Fig. 1.

  1. Table 1: in the footnote mention to which species each number corresponds

Response: We added the species each number corresponds in Table 1 footnote.

  1. Line 27 and line 28: please correct the name of the compounds.

Response: We have corrected the name of the compound.

  1. Lines 131, 133: Rosmarinus officinalis in italics

Response: We change Rosmarinus officinalis to italic.

  1. What were the reference standards used to determine the retention index? Please add them in the GC-MS analysis section.

Response: We added references in GC-MS analysis section.

  1. Why were essential oils not used in the enzyme activity assay? If the authors could make this determination, they might have more evidence to support their conclusions.

Response: We have considered using the essential oil to test the relevant enzyme activity reaction, but because the essential oil is a mixture, it is difficult to explain the specific problem. We have just tested the enzyme activity of the main compounds.

  1. In the discussion section, you could make evident the essential oil that showed the highest insecticidal activity and relate it to its chemical composition.

Response: In the discussion section, We added: To explore the relationship between the constituents of plant EOs and termiticidal activity, 8 main components were tested for insecticidal activity against R. dabieshanensis. In this study, the linalyl acetate, 1,8-cineole, thymol, carvone, citronellal, menthol, eugenol, and linalool display effective vapor activity against R. dabieshanensis, which are the major components of the nine selected EOs. In general, the insecticidal activity of the EOs may be attributed to its major component, have also been reported in some previous studies [Jin et al. 2022; Yang et al., 2021; Xie et al., 2019, 2020]. Here, the O. basilicum EO showed the highest insecticidal activity in comparison to its major constituents, eugenol against R. dabieshanensis. Similarly, Similarly, Piri et al. (2021) found the Ajwain EO showed the highest insecticidal activity in comparison to its constituents against Tuta absoluta larvae. Shahriari et al. (2017) reported that the Ajwain EO was more toxic to Ephestia kuehniella larvae than thymol. Results from this study suggested that the EOs exhibited the termiticidal activity can be attributed to their major active chemical constituents.

Round 2

Reviewer 3 Report

Dear authors, you attended to all the suggestions.

Author Response

Thank you very much for your comments regarding our manuscript. Those comments are very valuable and helpful for improving our paper.

Dear authors, you attended to all the suggestions.

Response: We appreciate your comments.